# In Vitro Anti-Rotaviral Activity of Bavachin Isolated from *Psoralea corylifolia* L. (*Fabaceae*)

**DOI:** 10.3390/vetsci11050188

**Published:** 2024-04-25

**Authors:** Jinseok Jung, Jaehoon Bae, Ji Sun Park, Seung Woong Lee, Jae-Ho Jeong, Su-Jin Park

**Affiliations:** 1Functional Biomaterial Research Center, Korea Research Institute of Bioscience and Biotechnology, 181 Ipsin-gil, Jeongeup-si 56212, Republic of Korea; honse5@kribb.re.kr (J.J.); baejae8@yahoo.co.jp (J.B.); jsp5928@kribb.re.kr (J.S.P.); lswdoc@kribb.re.kr (S.W.L.); 2Department of Microbiology, Chonnam National University Medical School, Hwasun 58128, Republic of Korea

**Keywords:** bavachin, *Psoralea corylifolia*, antiviral activity, rotavirus, viral replication

## Abstract

**Simple Summary:**

In this study, we examined how extracts and bavachin from *Psoralea corylifolia* affect bovine and porcine rotaviruses in a lab setting. We evaluated whether these substances could prevent the virus from attaching to cells and whether they could stop it from replicating. We found that, in the virucidal assay, the extracts and bavachin were unable to reduce viral infectivity by neutralizing the virus before it entered the host cells. However, when it was applied after the virus had already infected cells, bavachin showed strong antiviral effects. It notably reduced the virus’s ability to replicate, indicating its potential as a treatment for both bovine and porcine rotaviruses.

**Abstract:**

Rotavirus is the main causative agent of viral gastroenteritis among young animals worldwide. Currently, no clinically approved or effective antiviral drugs are available to combat rotavirus infections. Herein, we evaluated the anti-rotaviral activities of extracts and bavachin isolated from *Psoralea corylifolia* L. (*Fabaceae*) (*P. corylifolia*) against the bovine rotavirus G8P[7] and porcine rotavirus G5P[7] in vitro. Two assay strategies were performed: (1) a virucidal assay to reduce viral infectivity by virus neutralization and (2) a post-treatment assay to assess viral replication suppression. The results from the virucidal assay showed that the extracts and bavachin did not exert anti-rotaviral activities. In the follow-up analysis after treatment, bavachin exhibited robust antiviral efficacy, with 50% effective concentration (EC50) values of 10.6 μM (selectivity index [SI] = 2.38) against bovine rotavirus G8P[7] and 13.0 μM (SI = 1.94) against porcine rotavirus G5P[7]. Bavachin strongly suppressed viral RNA synthesis in the early (6 h) and late stages (18 h) after rotaviral infection. These findings strongly suggest that bavachin may have hindered the virions by effectively inhibiting the early stages of the virus replication cycle after rotaviral infection. Furthermore, confocal imaging showed that bavachin suppressed viral protein synthesis, notably that of the rotaviral protein (VP6). These results suggest that bavachin has strong antiviral activity against rotaviruses, inhibits viral replication, and is a candidate natural therapeutic drug targeting rotaviral infection. The utilization of bavachin isolated from *P. corylifolia* may contribute to decreased mortality rates, lower medication expenses, and enhanced economic viability in domestic farms.

## 1. Introduction

Rotavirus, belonging to the *Reoviridae* family, is the major cause of severe acute diarrhea in the neonates of many domestic animals, as well as the death of millions of young children in developing countries [1,2]. Among all infectious diseases, neonatal diarrhea is a significant concern; rotaviruses induce intense diarrhea in newborn and post-weaning piglets and calves [1,2]. Rotaviral gastroenteritis in the livestock industry is responsible for noteworthy economic losses attributed to mortality, diminished productivity, and escalated medication expenses [1,2]. It is a non-enveloped icosahedral virion consisting of triple-layered, double-capsid particles that encase the viral genome, composed of 11 segments of double-stranded RNA [1]. Rotaviruses manifest clinically with symptoms such as fever, vomiting, dehydration, and severe diarrhea, particularly affecting neonatal and post-weaning piglets and calves [1,2]. Although some vaccines have been developed for rotaviral diarrhea in piglets and calves, they are not easily available and provide incomplete protection [2,3]. Given the lack of effective vaccines against rotaviral diarrhea in domestic animals, there is a demand for antiviral drugs to treat rotaviral infections, particularly in situations where vaccination is not viable or when breakout rotaviral infections occur. Unfortunately, no approved or effective antiviral drugs are currently available for treating rotavirus infections, and treatment mainly depends on the replacement of fluids and electrolytes until recovery [3]. Consequently, the only measures available to control the disease are disinfection strategies designed to prevent viral entry into farms.

An alternative strategy utilizing medicinal plants has been proposed for treating rotavirus infections. The phytotherapeutic products derived from plant extracts are rich in unknown active compounds for the management of various diseases and cause minimal side effects compared with synthetic drugs [4,5]. The bioactive products derived from plants, such as *Achyrocline bogotensis* (Kunth) DC., Brazilian medicinal plants, *Quillaja saponaria*, and grapes, exhibit antiviral activity against human and animal rotaviruses [6,7,8,9]. These medicinal products are deemed ideal drug candidates due to their lower toxicity, fewer side effects, greater cost-effectiveness, and potential for eliciting a more effective response compared to commercially available anti-rotavirus agents [6,7,8,9]. However, natural compounds and medicinal plants are not readily accessible as treatments for rotavirus infections in humans or animals [10]. Nevertheless, diarrhea induced by rotaviruses persists uncontrollably; therefore, there is an urgent demand for new drugs to manage rotavirus infections. Bavachin is a natural bioactive flavonoid isolated from the seeds and fruits of *Psoralea corylifolia* L. (*Fabaceae*) (*P. corylifolia*). *P. corylifolia* is one of the most commonly used medicinal plants in traditional Asian medicine, including Korea, and its seeds have been used as a deobstruent, anthelmintic, and diuretic, as well as used for the treatment of vitiligo, stomach disorders, and certain skin diseases [11]. It contains multiple phytochemical compounds, including bakuchiol, psoralen, isoflavone, bavachromene, isobavachromene, bavachalcone, isobavachalcone, corylifol A, and bavachin [12,13]. Specifically, bavachin has several properties, including anti-platelet [14], anticancer [15], anti-inflammatory [16], immunoadjuvant [17], and antiviral [18] activities. Although various pharmacological activities have been associated with bavachin (Figure 1.) from *P. corylifolia*, its antiviral activity against rotavirus has not been reported. In this study, we evaluated the in vitro anti-rotaviral activity of extracts and bavachin isolated from *P. corylifolia*.

## 2. Materials and Methods

### 2.1. Materials

Dried *P. corylifolia* seeds (2.4 kg) were purchased from a local market. Methanol (MeOH, 10 L) was extracted for 7 days. The methanol extract was concentrated under a vacuum, resulting in the formation of a dark brown solid (214.5 g) via evaporation, which was suspended in distilled water (1 L) and extracted with ethyl acetate (EtOAc) solvent. Then, bavachin (a phenolic compound) (Figure 1) was isolated using an EtOAc extract [19]. The comprehensive procedure for compound purification has been outlined in a previous report [19].

### 2.2. Viruses and Cell Lines

MA-104 cells (African rhesus monkey kidney) were purchased from the ATCC CRL-2373.1 and maintained in Eagle’s minimum essential medium (EMEM) supplemented with 5% FBS, 100 µg/mL streptomycin, 100 U/mL penicillin, and 100 U/mL amphotericin B [20]. The 85A (porcine rotavirus, G5P[7]) and KJ-56 (bovine rotavirus, G8P[7]) rotaviruses were isolated from fecal samples of diarrheic Korean piglets and calves. The rotaviruses were pre-activated with 10 µg/mL trypsin (1:250) procured from the GIBCO Invitrogen Corporation, California for 30 min at 37 °C before inoculation into MA-104 cells (confluent) [10]. Afterward, the infected MA-104 cells were cultured in a 1 µg/mL trypsin-containing medium.

### 2.3. Cytotoxicity

The MA-104 cells were cultured in 96-well plates (1 × 10^4^ cells/well) at 37 °C in a 5% CO_2_ atmosphere for 24 h (90% confluence). The medium was replaced with fresh media containing a serially diluted MeOH or EtOAc extract (1–100 µg/mL) or bavachin (1–100 µM) dissolved in dimethyl sulfoxide (DMSO) (with a final concentration at 0.5% DMSO), and the cells were cultured for 72 h. The control groups (untreated cells) were cultured in the presence of 0.5% DMSO. The solution was replaced with fresh media, and a 10 µL MTT (3-(4,5-dimethylthiozol-2-yl)-3,5-dipheryl tetrazolium bromide, Sigma, St. Louis, MO, USA) solution was added to each well and incubated at 37 °C for 4 h. Then, the supernatant was removed, and 100 µL of 0.04 M HCl-isopropanol was added to solubilize the formazan crystals. Absorbance at 540 and 655 nm was measured using a microplate reader. The 50% cytotoxic concentration (CC_50_) was calculated using regression analysis.

### 2.4. Antiviral Assay

The antiviral assays were performed as described in a previous study [21]. The results were visualized as previously described (the neutral red method). Briefly, in the virucidal assay, the extracts or bavachin (molecular weight: 324.4) was combined with varying concentrations (1–20 µg/mL or µM) with a 0.01 multiplicity of infection (MOI) rotavirus and incubated at 4 °C for 1 h. Then, the mixture of extracts or bavachin and the rotavirus was inoculated into MA-104 cell monolayers (confluent, 1 × 10^4^ cells/well) and incubated for 1 h and occasionally shook. The solution was removed and replaced with EMEM containing 1 µg/mL trypsin with a final concentration at 0.5% DMSO. Subsequently, the cells were incubated for 72 h at 37 °C under a 5% CO_2_ atmosphere until the infected/untreated cells exhibited complete viral cytopathic effects (CPEs), which were assessed via light microscopy. In the post-treatment assay, 0.01 MOI rotaviruses were inoculated in 1 × 10^4^ cells/well MA-104 cell monolayers (nearly confluent) and incubated for 1 h with occasional shaking. The solution was then removed and replaced with EMEM containing 1 µg/mL trypsin and 1–20 µg/mL or µM of each extract or bavachin (final concentration of 0.5% DMSO). Infected/untreated and uninfected/untreated cells were cultured in the presence of 0.5% DMSO. Subsequently, the cells were incubated for 72 h at 37 °C until the infected/untreated cells exhibited complete viral cytopathic effects (CPEs). Then, a neutral red solution was added to each well at a concentration of 0.034% (*w*/*v*) and incubated for 2 h at 37 °C (in the dark). Following the incubation period, the neutral red solution was removed, and the MA-104 cells were gently washed with phosphate-buffered saline (PBS, pH 7.4), and a destaining solution composed of 1% glacial acetic acid, 49% water, and 50% ethanol was added. The plates were incubated in the dark (15 min at room temperature). Then, the absorbance of each well was measured using a microplate reader (540 nm). A 50% effective concentration (EC_50_) was determined using regression analysis. To assess the selectivity of the compounds, the selectivity index (SI) was computed using the following formula: SI = CC_50_/EC_50_.

### 2.5. Reverse Transcription and Quantitative Real-Time PCR (qRT-PCR)

The MA-104 cells were cultured to confluency at 37 °C in a 5% CO_2_ atmosphere for 24 h. To assess the dose-dependent inhibition of bavachin, the MA-104 cells infected with rotavirus at 0.01 MOI were maintained with bavachin (a final concentration at 0.5% DMSO) at different concentrations (5–15 µΜ) for 24 h. To assess the inhibitory activity of bavachin in the early and late replication steps, the confluent MA-104 cells infected with rotavirus (0.01 MOI) were cultured with 15 µΜ bavachin for 6 h and 18 h. The infected/untreated and uninfected/untreated cells were cultured with a final concentration at 0.5% DMSO. After incubation for 6 h, 18 h, and 24 h, the culture medium was removed, and the cells were scraped off, washed twice with PBS, and collected using centrifugation at 500× *g* for 3 min. The total RNA was extracted using the Qiagen RNeasy Mini Kit (QIAGEN, Hilden, Germany) according to the manufacturer’s guidelines. The cDNA was synthesized from the total RNA using the cDNA Master Mix (random primer) (Applied Biosystems, Foster City, CA, USA). Specific primers were used to facilitate the quantitative real-time PCR of synthetic cDNA (NSP 3 gene, 5′-ACCATCTWCACRTRACCCTCTATGAG-3′ [upstream] and 5′-GGTCAC ATAACGCCCCTATAGC-3′ [downstream]). Glyceraldehyde-3-phosphate dehydrogenase (GAPDH) was used as an internal control for cellular RNAs and amplified using the primer sequences 5′-TCAACAGCGACACCCACTC-3′ (upstream) and 5′-CTTCCTCTTGTGCTCTTGCTG-3′ (downstream). qRT-PCR was performed using the Step One Plus Real-time PCR System using 2 µL cDNA and the Power SYBR Green PCR 2X Master Mix (Applied Biosystems, CA, USA). Data analysis was performed using the v2.1 StepOne software from Applied Biosystems, Foster City, CA, USA.

### 2.6. Confocal Fluorescence Imaging

The MA-104 cells were cultured to confluency at 37 °C in a 5% CO_2_ atmosphere for 24 h. The MA-104 cells were infected with the rotaviruses (G8P[7] bovine and G5P[7] porcine) at a 0.01 multiplicity of infection (MOI) for 1 h and occasionally shook. The solution was removed and replaced with α-MEM, containing 1 µg/mL trypsin and 15 µM (a final concentration at 0.5% DMSO) of the compound. The infected/untreated and uninfected/untreated cells were cultured with a final concentration of 0.5% DMSO. All cells were then incubated for 24 h at 37 °C in a 5% CO_2_ atmosphere, gently washed three times with pH 7.4 PBS, and fixed in 4% paraformaldehyde solution (15 min at room temperature). After three additional washes with pH 7.4 PBS, the cells were incubated with a monoclonal antibody targeting the rotavirus VP6 protein from Santa Cruz (Dallas, TX, USA) diluted (1:50 in pH 7.4 PBS) at 37 °C for 1 h. Then, the cells were washed with pH 7.4 PBS and incubated with a FITC-conjugated goat anti-mouse IgG antibody diluted at 1:100 in pH 7.4 PBSat 37 °C for 1 h. The cells were washed with pH 7.4 PBS, stained with 500 nM propidium iodide (PI) solution for 10 min (room temperature), and washed three times with pH 8.0 PBS. The slides were mounted using the SlowFade Gold antifade reagent from Invitrogen, Carlsbad, CA, USA. Confocal fluorescence imaging was performed using the Carl Zeiss LSM 510 META confocal microscope from Carl Zeiss Inc., Jena, Germany. The quantification of confocal fluorescence images was performed using ImageJ software (version 1.8.0).

### 2.7. Statistical Analysis

The experiments were conducted in triplicate to ensure replicability. The data obtained are expressed as the mean ± standard error (SE) to indicate the central tendency and variability, respectively. Statistical analysis was conducted using the Sigma Plot Statistical Analysis software version 14.0. The differences between the group mean values were determined using a one-way analysis of variance, followed by the Tukey’s test, assuming equal variances.

## 3. Results

### 3.1. Cytotoxicity of the Extracts and Bavachin from P. corylifolia in MA-104 Cells

In this study, we confirmed whether the extracts or isolated bavachin from *P. corylifolia* possess inhibitory activity against rotavirus. Figure 1 shows the structure of bavachin isolated from *P. corylifolia*. The cytotoxicity of the MeOH and EtOAc extracts and bavachin was evaluated by determining the CC_50_ using the MTT assay. The CC_50_ of the MeOH and EtOAc extracts were 24.4 and 28.5 µg/mL, respectively, while that of bavachin was 25.2 μM (Table 1). The concentrations of bavachin below 15 μM did not exhibit any toxic effects (cell viability ≥ 95%) against the MA-104 cells. To evaluate its antiviral effects, bavachin at concentrations with minimal toxicity were used in the subsequent experiments.

### 3.2. Antiviral Activity of Bavachin Isolated from P. corylifolia against Rotavirus

To investigate the specific stage at which bavachin exerted its inhibitory effects, time-of-addition experiments were conducted, including a concurrent treatment during viral infection (virucidal assay) and post-infection (post-treatment assay) [13]. To evaluate the ability of bavachin to prevent rotavirus attachment to the MA-104 cells, a simultaneous-treatment experimental paradigm was utilized. Our results indicated that bavachin did not exert any antiviral activity for rotaviral neutralization (data not shown).

To further assess the inhibitory effect of bavachin on viral replication, post-treatment assays were performed. The rotavirus-uninfected/untreated cells served as the normal control, and the rotavirus-infected/untreated cells served as the viral-infected group. The rotavirus-infected/untreated cells showed CPEs in the MA-104 cells; by contrast, the control group exhibited no morphological changes. The MeOH extract showed an antiviral effect against 85A (G5P[7], porcine rotavirus) (EC_50_ = 12.4 μg/mL, SI = 2.05), but no antiviral activity was observed in KJ-56 (G8P[7], bovine rotavirus) (Table 1). Also, the EtOAc extract showed an antiviral effect against KJ-56 (G8P[7], bovine rotavirus) (EC_50_ = 13.5 μg/mL, SI = 2.14) and 85A (G5P[7], porcine rotavirus) (EC_50_ = 14.0 μg/mL, SI = 2.06) (Table 1). Bavachin significantly demonstrated an anti-rotaviral effect in a dose-dependent manner (1–20 µM) against KJ-56 (G8P[7]) and 85A (G5P[7]). The results showed that bavachin inhibited KJ-56 (G8P[7]) (EC_50_ = 10.6 μM, SI = 2.37) and 85A (G5P[7]) (EC_50_ = 13.0 µM, SI = 1.94) (Table 1). Collectively, these results confirmed that bavachin exhibits antiviral activity by inhibiting viral replication after rotavirus infection.

### 3.3. Inhibition of Viral RNA and Viral Protein Synthesis during Rotavirus Replication

To examine the inhibitory effects of bavachin on viral RNA synthesis, the levels of intracellular viral RNA were assessed in the infected cells after treatment with different concentrations of bavachin (5–15 μM) with a final concentration at 0.5% DMSO. The total RNA was extracted from the infected cells 24 h after rotavirus infection, and quantitative real-time RT-PCR was conducted using specific primers targeting the viral NSP3 gene. As shown in Figure 2A, bavachin significantly reduced the viral RNA levels in a dose-dependent manner. In the presence of 10 µΜ bavachin, the viral RNA levels decreased by 70% more compared to those in the mock-treated cells (Figure 2A). Furthermore, the inhibitory effect of bavachin (15 μM) on RNA synthesis in the early and late stages (6 h and 18 h) of rotavirus infection in the MA-104 cells is discussed. As shown in Figure 2B, both the viral RNA levels in the cells treated with bavachin were markedly decreased by 97–98.3% in the early stage. In addition, both the viral RNA levels in the cells treated with bavachin were markedly decreased by 91.2–93.4% in the late stage compared with those of the untreated cells (Figure 2B). These results clearly show that bavachin may have inactivated the virions directly or indirectly by effectively inhibiting the early-stage rotavirus replication cycle after viral infection.

Remarkably, we investigated the intracellular trafficking of the rotavirus VP6 protein in virus-infected cells using immunofluorescence analysis. The virus-infected cells exhibited green fluorescence, providing visual evidence of rotavirus replication (Figure 3B,E), while the mock-infected MA-104 cells did not (Figure 3A,D). However, when the cells were treated with 15 μM bavachin (Figure 3C,F) after rotavirus infection, a noticeable reduction in the number of positive cells (fluorescence) was observed, indicating that bavachin disrupted viral protein production (Figure 3G). The results from confocal fluorescence imaging also confirmed that bavachin suppressed the amount of rotavirus virions in the host cells. Overall, these results supported the hypothesis that bavachin inhibits viral replication by inhibiting viral RNA and protein expression.

## 4. Discussion

Products from natural sources, mainly plants, have been utilized for food preservation and have fulfilled the primary healthcare needs in every known culture. Furthermore, there is a growing interest in exploring the potential antiviral efficacy of medicinal plants in both experimental and clinical settings, particularly in targeting various stages of rotavirus infection [8,22,23,24]. Compounds and extracts isolated from *P. corylifolia* seeds have been reported to have antiviral effects against SARS-CoV PLpro, influenza virus (pheolic isopremoid), and porcine reproductive and respiratory syndrome virus [25,26,27]. However, the anti-rotaviral activity of bavachin remains unclear. This study was conducted to determine the antiviral activity of *P. croylifolia* extracts and bavachin against rotavirus.

The early-stage rotavirus replication cycle has been reported to involve cell attachment and entry, the synthesis and translation of viral mRNAs, and the formation of viroplasms. The late stages of infection are associated with viral particle maturation and viral release from the host cells [1,28]. Anti-rotavirus agents target these various stages in the rotaviral replication cycle [1,28]. We hypothesized that the MeOH and EtOAc extracts and bavachin isolated from *P. corylifolia* inhibits the viral–cell binding process via virus neutralization during viral entry and/or inhibits the expression of viral RNA and proteins after entry into the cells. Time-of-addition experiments were performed to ascertain the stage at which bavachin exerts its inhibitory activity. The two extracts and bavachin were incubated with MA-104 cells at two distinct time points: after incubation for 1 h at 4 °C with the virus prior to virus infection using virucidal assay, and 1 h after virus inoculation (post-treatment assay). As a result, the MeOH extract exhibited antiviral activity only against the porcine rotavirus G5P[7] only in the post-treatment assay. But the results showed that the EtOAc extract and bavachin exerted their inhibitory effects against both viruses (the bovine rotavirus G8P[7] and the porcine rotavirus G5P[7]) only in the post-treatment assay.

Given its established ability to inhibit rotavirus-induced cytopathic effects (CPEs) in the post-treatment assay, we proceeded to investigate the inhibitory effects of bavachin on rotaviral RNA synthesis and rotaviral protein expression. As a result, bavachin inhibited rotavirus RNA synthesis in a dose-dependent manner (Figure 2A). Also, Urasawa et al. reported that the virus-infected cells were observed in the cytoplasm after 6 h, and cell-to-cell infection appeared after 18 h [29]. The detachment of cells from the monolayer cell sheet became remarkable after 24 h [29]. Therefore, in this study, we designed an experiment, denoting the early stage as 6 h after rotavirus infection and the late stage as 18 h. Consequently, bavachin predominantly reduced the viral RNA levels in the early and late stages (6 h and 18 h) (Figure 2B). These findings strongly suggest that bavachin could have inhibited the virions by efficiently impeding the early phases of the rotavirus replication cycle after viral infection. Confocal microscopy also showed that bavachin dramatically suppressed VP6 synthesis in the MA-104 cells (Figure 3). It was also confirmed that bavachin reduced the amount of rotavirus virions in the MA-104 cells. Nevertheless, these results cannot exclude the possibility that bavachin inhibits the cellular mechanisms utilized during viral replication. This is an interesting subject for follow-up studies.

Viruses depend on broad adaptations to the cellular biosynthetic machinery of host cells to facilitate their replication because of their small genome size [9]. Also, rotavirus hijacks and depends on the host cellular signaling pathways during its life cycle, especially those involved in apoptosis and cell survival [30,31]. Among these, the PI3K/Akt and mitogen-activated protein kinase (MAPK) signaling pathways have attracted considerable interest because of their multifaceted roles in modulating viral entry, replication, assembly, and release [32,33]. Rotavirus infection induces the activation of the transcription factor AP-1 via the MAPK signaling pathway in the human intestinal epithelial cell line (Caco2) [34]. Furthermore, the activation of ERK 1/2 and p38 plays an important role in rotavirus replication [35]. Interestingly, bavachin has been reported to effectively suppress the LPS-induced activation of MAPK signaling pathways (JNK 1/2 and ERK 1/2) [16]. Consequently, we are currently attempting to elucidate the precise mechanisms by which bavachin exerts its effects on rotavirus replication.

## 5. Conclusions

The findings of this study demonstrated the potential therapeutic effectiveness of bavachin derived from *P. corylifolia* in combating rotavirus infections in vitro. In conclusion, our study showed that bavachin isolated from *P. corylifolia* exerted antiviral activity against rotavirus infections (both G8P[7] bovine rotavirus and G5P[7] porcine rotavirus) by inhibiting viral RNA synthesis and protein expression in the host cells. These findings suggest that the broad-spectrum antiviral activity of bavachin highlights its potential as a candidate for the development of effective antiviral therapies. Nonetheless, it is crucial to recognize that further research is necessary not only to clarify the application of bavachin but also to understand its role in the identified antiviral effects. These subsequent studies will contribute to our understanding of the therapeutic potential and action mechanism of bavachin concerning rotavirus infections.

## Figures and Tables

**Figure 1 vetsci-11-00188-f001:**
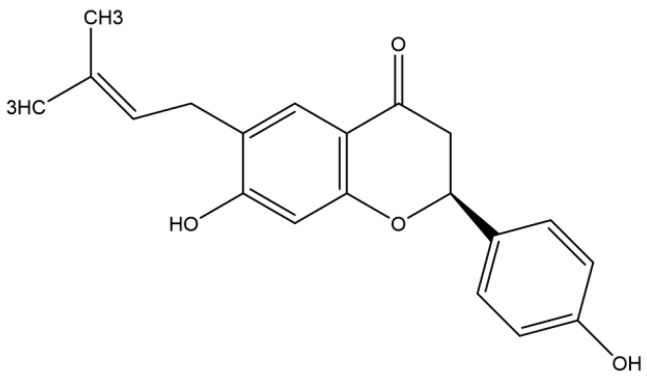
Structure of bavachin isolated from *Psoralea corylifolia*.

**Figure 2 vetsci-11-00188-f002:**
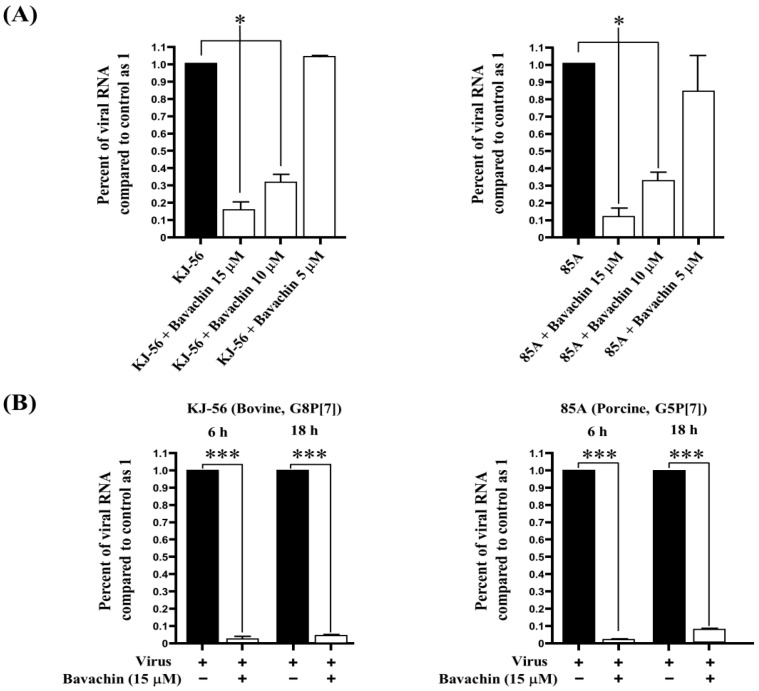
Suppression of rotavirus viral RNA synthesis (NSP3 gene) through bavachin treatment. (**A**) MA-104 cells were infected with rotavirus (G8P[7] bovine rotavirus and G5P[7] porcine rotavirus) at 0.01 multiplicity of infection (MOI). After 1 h, viruses were removed, and cells were then subjected to treatment with either 0.5% dimethyl sulfoxide (DMSO) or bavachin (5–15 μM). After 24 h of incubation, total RNA was extracted, and intracellular viral RNA levels were quantified. Rotavirus RNA levels were normalized using GAPDH. (**B**) MA-104 cells were infected with the 0.01 MOI rotaviruses. After 1 h, rotaviruses were removed, and the solution replaced with α-MEM supplemented with 0.5% DMSO or 15 μM of bavachin. Total RNA was extracted at 6 h and 18 h after infection, and levels of intracellular rotavirus RNA were quantified. Rotavirus RNA levels were normalized using GAPDH. Data are presented as mean ± SEM (n = 3), * *p* < 0.05, *** *p* < 0.001.

**Figure 3 vetsci-11-00188-f003:**
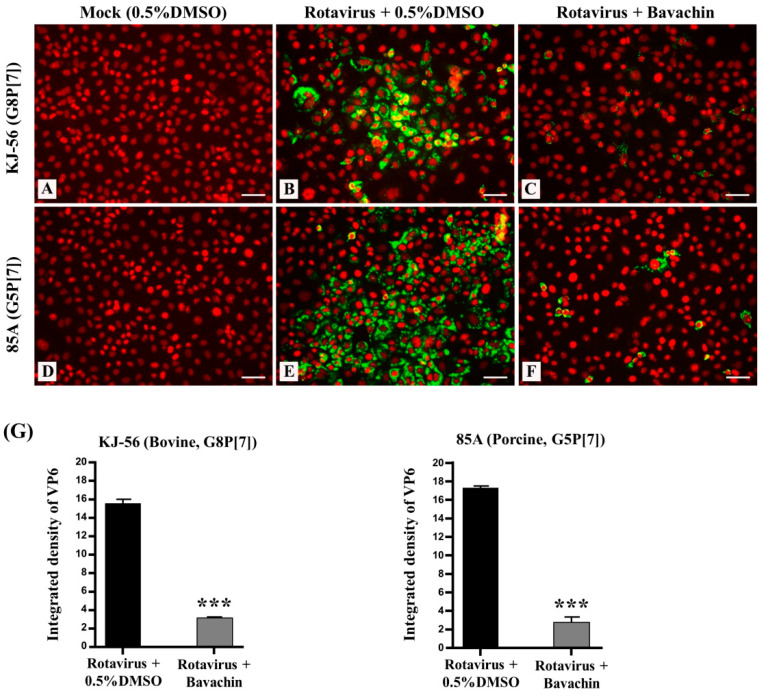
Suppression of rotavirus viral protein synthesis (VP6) through bavachin treatment. MA-104 cells were infected with rotaviruses (bovine rotavirus, G8P[7] and porcine rotavirus, G5P[7]) at 0.01 multiplicity of infection (MOI) for 1 h. The solution was removed and replaced with α-MEM supplemented with (**B**,**E**) 0.5% DMSO or (**C**,**F**) 15 μM of bavachin (**C**,**F**), or (**A**,**D**) cells were mock-infected. After 24 h, cells were fixed in 4% paraformaldehyde, blocked, and incubated with an anti-VP6 antibody (green). Propidium iodide was used as nuclear counterstain (red). Scale bar = 50 μm. (**G**) Quantification of green fluorescence related to VP6 protein levels was performed using version 1.8.0. ImageJ software. Data are presented as mean ± SEM (*n* = 3), *** *p* < 0.001.

**Table 1 vetsci-11-00188-t001:** In vitro anti-rotavirus activities of MeOH and EtOAc extracts and bavachin isolated from *Psoralea corylifolia* against KJ-56 (bovine rotavirus, G8P[7]) and 85A (porcine rotavirus, G5P[7]) on MA-104 cells using a post-treatment assay. ^a^ CC_50_ mean (50%) value of cytotoxic concentration. ^b^ EC_50_ mean (50%) value of effective concentration. ^c^ SI selective index, CC_50_/EC_50_.

Extracts or Compounds		CPE Reduction Assay			
	CC_50_ ^a^	KJ-56 (G8P[7])		85A (G5P[7])	
		EC_50_ ^b^	SI ^c^	EC_50_ ^b^	SI ^c^
MeOH extract	25.4 ± 7.0 μg/mL	-	-	12.4 ± 1.9 μg/mL	2.05
EtOAc extract	28.9 ± 5.7 μg/mL	13.5 ± 4.0 μg/mL	2.14	14.0 ± 0.4 μg/mL	2.06
Bavachin	25.2 ± 1.2 μM	10.6 ± 0.3 μM	2.38	13.0 ± 4.9 μM	1.94

## Data Availability

The data used in this current study are available from the corresponding author upon request.

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
