# Peer review of "In Vitro Anti-Rotaviral Activity of Bavachin Isolated from Psoralea corylifolia L. (Fabaceae)"

_vetsci, 2024, doi:10.3390/vetsci11050188_

Round 1

Reviewer 1 Report (Previous Reviewer 1)

Comments and Suggestions for Authors

The manuscript has been improved. However, several concerns must be further addressed.

Major concern:

1.       Line 164: There are many cDNA Master Mix kits. A specific one should be mentioned.  Only one primer is required for the RT reaction. Why do you use two primers for the RT?

2.       Bavachinin is the only active compound in MeOH and EtOAc extracts, which should be presented with references.

3.       In Table 1, extract that does not carry activity has to be indicated. Further discussion is required for this absence of activity.

4.       Line 230: negative result should also be presented.

5.       The rationale of treating virus with compounds at 6h and 18h should be presented, as found in the response 10 of the reviewer.

6.       Line 270: “infection” should be used instead of “replication” because the reduction in viral RNA and protein expression may not mainly result from inhibiting the viral replication but other factors.

Comments on the Quality of English Language

I don't have any further comments.

Author Response

  1. Line 164: There are many cDNA Master Mix kits. A specific one should be mentioned.  Only one primer is required for the RT reaction. Why do you use two primers for the RT?

Response 1: Thank you for your comment. As your comments, we made a mistake: we used random primers for the RT reaction and specific primers for the quantitative real-time PCR. Thus, we corrected it in the manuscript at line 161-163.

  1. Bavachin is the only active compound in MeOH and EtOAc extracts, which should be presented with references.

Response 2: Thank you for your comment. We have already cited the references in this manuscript (ref. 19).

  1. In Table 1, extract that does not carry activity has to be indicated. Further discussion is required for this absence of activity.

Response 3: Thank you for your helpful comment. According to your comment, we corrected the manuscript at line 303-313.

  1. Line 230: negative result should also be presented.

Response 4: Thank you for your suggestion. As your comment, we corrected the manuscript at line 230-231.

  1. The rationale of treating virus with compounds at 6h and 18h should be presented, as found in the response 10 of the reviewer 2.

Response 5: Thank you for your comment. Following your comment, we have made corrections to Figure 2B, comparing viral RNA levels after treatment to those of untreated viral RNA controls at both 6 hours and 18 hours.

  1. Line 270: “infection” should be used instead of “replication” because the reduction in viral RNA and protein expression may not mainly result from inhibiting the viral replication but other factors.

Response 6: Thank you for your comment. Generally, rotavirus replication cycles were reported to involve early stages such as cell attachment and entry, synthesis and translation of viral mRNAs, and formation of viroplasms. The later stages of infection are associated with viral particle maturation and viral release from host cells [Trejo-Cerro et al., 2018 (ref. 28)]. As your comment, although there could be various reasons for the reduction of viral RNA and protein expression, ultimately, all of these contribute to the inhibition of the virus replication process. Therefore, we are currently attempting to determine the specific mechanisms through which the compound act against rotavirus.

Trejo-Cerro, Ó.; Eichwald, C.; Schraner, E.M.; Silva-Ayala, D.; López, S.; Arias, C.F. Actin-Dependent Nonlytic Rotavirus Exit and Infectious Virus Morphogenetic Pathway in Nonpolarized Cells. Journal of Virology 2018, 92(6), e02076-17.

Reviewer 2 Report (New Reviewer)

Comments and Suggestions for Authors

 Use Italic for Latin names (Fabaceae and so on)

Lines 111, 130 and 135  MA-104 cells were seeded 1×10^5 cells/well in 96-well plates”. According to the recommendations 1×10^4 cells/well in 96-well plates should be seeded. Check the reference and explain:

Detection and Quantification of African Swine Fever Virus in MA-104 Cells (bio-protocol.org)

Lines 127-128 “extracts or bavachin were mixed with varying concentrations (1–20  µg/mL or µM)”. The concentrations expressed as 1 µg/ml does not mean that it will be equal to 1 µM. What about the molecular weight of bavachin? You should write the molecular weight.

 Line 128 -  It is written that “0.01 multiplicity of infection (MOI)” was chosen. I doubt that the correct number of cells was used for the calculation of MOI. Please, check and confirm that is true. MOI is the ratio of number of virus and cells (virus/cells). If you had chosen incorrectly 10 times higher number of cells for MOI calculation, the ratio will be 10 times lower. What was the titre (TCID50) of viral strains? How many viruses (the measurement by TCID50 or other method) were used per one well of 96-well plate? If TCID50 assay was used, was 1 TCID50 regarded as one virus?

Line 128 Why was incubation at 4 °C for one hour chosen? Why not room temperature? What about bavachin molar concentration in both extracts?

Line 252 “viral RNA levels showed a 30% decrease compared to those in the mock-treated” What do RNA levels mean? How was 30% decrease of viral RNA levels calculated? Is it the result or the speculation (interpretation of results) according to the Ct values. Chapter 2.5. (Reverse transcription and quantitative real-time PCR (qRT-PCR)) does not explain it.

Lines 167, 277, 281 GAPDH – there is no explanation of abbreviation.

Figure1 should be included into Introduction, but not the Results

Figure 2 Check the units in the scale from 0 to 1.1 mean.

Figure 2 When you compare the suppression of rotavirus viral RNA synthesis (NSP3 gene) through bavachin treatment after 6 and 18 hours you should use two controls or should not you?

1.       Untreated Viral RNA control after 6 hours vs viral RNA after treatment

2.       Untreated Viral RNA control after 18 hours vs viral RNA after treatment

The conclusions are devoted to bavachin only. What about efficacy of extracts?

The conclusion “P. corylifolia exerted antiviral activity against rotavirus infections (both G8P[7] bovine rotavirus and G5P[7] porcine rotavirus) by inhibiting viral RNA synthesis and protein expression in host cells ” should be revised. Only one type cells were used.

Comments on the Quality of English Language

UK English native speaker-grammar specialist  must review the publication and confirm it. 

Author Response

Use Italic for Latin names (Fabaceae and so on)

 Response : Thank you for your suggestion. We corrected the manuscript.

  1. Lines 111, 130 and 135 : MA-104 cells were seeded 1×10^5 cells/well in 96-well plates”. According to the recommendations 1×10^4 cells/well in 96-well plates should be seeded. Check the reference and explain: Detection and Quantification of African Swine Fever Virus in MA-104 Cells (bio-protocol.org)

Response 1: Thank you for your suggestions. According to your comments, we checked cell number. So, we made a mistake and corrected it in the manuscript at line 109, 127, and 133.

  1. Lines 127-128: “extracts or bavachin were mixed with varying concentrations (1–20  µg/mL or µM)”. The concentrations expressed as 1 µg/ml does not mean that it will be equal to 1 µM. What about the molecular weight of bavachin? You should write the molecular weight.

Response 2: In accordance with your suggestion, we added the (molecular weight: 324.4) of bavachin in the manuscript at line 124.

  1. Line 128: It is written that “0.01 multiplicity of infection (MOI)” was chosen. I doubt that the correct number of cells was used for the calculation of MOI. Please, check and confirm that is true. MOI is the ratio of number of virus and cells (virus/cells). If you had chosen incorrectly 10 times higher number of cells for MOI calculation, the ratio will be 10 times lower. What was the titre (TCID50) of viral strains? How many viruses (the measurement by TCID50 or other method) were used per one well of 96-well plate? If TCID50 assay was used, was 1 TCID50 regarded as one virus?

Response 3: Thank you for your comment. According to your comment, we checked virus titers. The viruses we used in the experiment, the KJ-56 strain (bovine rotavirus, G8P[7]) and the 85A strain (porcine rotavirus, G5P[7]), showed TCID50 9.8 X 10^7 PFU/mL and 4.1 X 10^7 PFU/mL, respectively. Hence, it is correct that we utilized a virus amount corresponding to a MOI of 0.01 in the experiment.

  1. Line 128: Why was incubation at 4 °C for one hour chosen? Why not room temperature? What about bavachin molar concentration in both extracts?

Response 4: Thank you for your comment. The virucidal assay used in this manuscript is conducted to determine whether compounds directly inhibit the virus [Park et al. 2023 (ref.10)]. Consequently, a one-hour incubation period at 4°C is provided to allow for the interaction between the compound and the virus without altering the viral titer.

  1. Line 252 “viral RNA levels showed a 30% decrease compared to those in the mock-treated” What do RNA levels mean? How was 30% decrease of viral RNA levels calculated? Is it the result or the speculation (interpretation of results) according to the Ct values. Chapter 2.5. (Reverse transcription and quantitative real-time PCR (qRT-PCR)) does not explain it.

Response 5: Thank you for your suggestion, we made a mistake and amended it in the manuscript at line 247-248.

  1. Lines 167, 277, 281 GAPDH – there is no explanation of abbreviation.

Response 6: Thank you for your suggestion. As your comment, we corrected it in the manuscript at line 164-165.

  1. Figure1 should be included into Introduction, but not the Results

Response 7: Thank you for your comment. As your comment, we corrected the manuscript at line 83 and 93.

  1. Figure 2 Check the units in the scale from 0 to 1.1 mean.

Response 8: Thank you for your comment. According to your comment, we checked Figure 2 but found no issues.

  1. Figure 2 When you compare the suppression of rotavirus viral RNA synthesis (NSP3 gene) through bavachin treatment after 6 and 18 hours you should use two controls or should not you? (Untreated Viral RNA control after 6 hours vs viral RNA after treatment/Untreated Viral RNA control after 18 hours vs viral RNA after treatment).

Response 9: Thank you for your helpful comment. Following your comment, we have made corrections to Figure 2B, comparing viral RNA levels after treatment to those of untreated viral RNA controls at both 6 hours and 18 hours.

  1. The conclusions are devoted to bavachin only. What about efficacy of extracts?

Response 10: Thank you for your comment. Since bavachin was isolated from the extracts, we have confirmed the anti-rotavirus activity of the extracts in this manuscript. These results have been documented in the results and discussion sections of this manuscript at line 229-233 and 310-313.

  1. The conclusion “ corylifoliaexerted antiviral activity against rotavirus infections (both G8P[7] bovine rotavirus and G5P[7] porcine rotavirus) by inhibiting viral RNA synthesis and protein expression in host cells ” should be revised. Only one type cells were used.

Response 11: Thank you for your comment. The propagation of rotaviruses is generally performed in several cell types of kidney or colon adenocarcinoma (Arnold et al., 2009). MA-104 cells (Rhesus monkey kidney cell line) in addition to CaCo-2 (Human colon carcinoma cell line) and HepG2 cells (Human hepatocellular liver carcinoma cell line) are the cells of choice for rotavirus growth (Otto et al, 2015). It was reported that rotaviruses from of different animal species were submitted for isolation and cultural propagation of rotavirus on MA-104 cells (Otto et al, 2015). Also, this paper showed that MA-104 cells were successfully used not only for the initial adaption of several rotavirus strains, but also for their continuous propagation in vitro (Otto et al, 2015).

Arnold, M.; Patton, J.T.; McDonald, S.M. Culturing, storage, and quantification of rotaviruses. Current Protocols in Microbiology 2009. 15C.3.1–15C.3.24.

Otto, P.H.; Reetz, J.; Eichhorn, W.; Herbst, W.; Elschner, M.C. Isolation and propagation of the animal rotaviruses in MA-104 cells--30 years of practical experience. Journal of Virological Methods 2015, 223:88-95.

Round 2

Reviewer 1 Report (Previous Reviewer 1)

Comments and Suggestions for Authors

The authors have already addressed all the concerns. I don't have any further comments.

Comments on the Quality of English Language

I don't have any further comments.

This manuscript is a resubmission of an earlier submission. The following is a list of the peer review reports and author responses from that submission.

Round 1

Reviewer 1 Report

Comments and Suggestions for Authors

Rotavirus is a non-envelope virus and leads to severe diarrhea in children and animals. In this study, Jung et al and his colleagues demonstrated that bavachin isolated from Psoralea corylifolia L. could be the potential inhibitor of rotavirus in vitro.  However, some major concerns must be addressed.

Major comment:

1.    The control in the determination of the extract potency and toxicity was improperly used. There were two extracts. One from MeOH is suspended in water finally, while the solvent to resuspend was not clearly described. Corresponding solvent is required to be used as the negative/vehicle control instead of using DMSO alone.

2.    The method of the antiviral assay was not clearly described. Line 115-116, what temperature was used for 1 h incubation? In the simultaneous treatment, virus was incubated with the compound for 1h at 4dC. Was the compound absent during the subsequent 72h incubation? In the simultaneous treatment, the compound was incubated with the virus in the absence of trypsin; in the post-treatment assay, the virus was incubated with the virus-infected cells and trypsin. The presence of the trypsin seems to be critical. There are two variables between these assays, which would make the results not conclusive.

3.    The method of the rt-qPCR was not clearly described. The primers for qPCR were not clearly described. Another critical concern is about the reverse transcription. In principle, only one primer is required for the reaction. Why do you use two primers? Which cDNA kit was used?

4.    The method of the rt-qPCR was not clearly described. What is the infection condition? How do you permeabilize the cells for antibody staining?

5.    The MeOH and EtOAc extracts should include other organic molecules. It is not appropriate to compare their results to the pure bavachin. Further characterization should be required to demonstrate that bavachin is the major component in those extracts.

6.    Line 186: the description of the virucidal assay was absent.

7.    In Table 1, why was EC50 of MeOH against KJ-56 absent? SI unit does not have any units.

8.    Line 197, result was absent to support the statement.

9.    In the simultaneous treatment, the compound did not show any effects on the viral attachment. Statement on line 202 over-claimed the compound’s effect.

10. In line 215, early and late stages of rotavirus infection were stated. Further definition is required. How long does one replication cycle of rotavirus take? At 18h, the virus should have gone through 1.5 replication cycle.

11. Line 228, Statement on line 202 over-claimed the compound’s effect. The reduction in viral RNA and protein expression may not mainly result from inhibiting the viral replication but delaying the virus entry or forcing virus to the lysosomes and so on.

12. References are required for certain statements, line 55-60, line 61-62, line 85-88,

Comments on the Quality of English Language

Careful proofreading is required. There are lots of grammatical and typo-errors.

Reviewer 2 Report

Comments and Suggestions for Authors

Comments to the author:

In this study, the authors showed the neutralization against rotavirus by extracts and bavachin from Psoralea corylifolia. They found that bavachin exhibited its antiviral activity by suppressing the virus replication rather than preventing the virus from attaching to cells.

The experimental design is not very rigorous. I don’t recommend it for publication unless the author can fully complete the suggested experiments and respond to my comments.  

Please find below my comments:

Major marks:

1. Was bavachin used in this study purchased from company or made in-house by the author? The author should declare it in the manuscript. If made in-house, it will be better that the author provide the purity and other relevant data to show the profile of purified bavachin. If commercial bavachin, please indicate the vendor and catalog number.

2. This study was mainly focused on bavachin. What is the purpose and value of involving “MeOH” and ”EtOAc” in this work? How did the author confirm that the extracts from dried of P. corylifolia seeds are bavachin?

3. What is the replicative cycle for ratovirus? How long does it take?  How can author define the early stage and late stage for viral replication, is there any reference? What are the differences and activities during early and late stages?

4. Regarding the “3.3. Inhibition of viral RNA and viral protein synthesis during rotavirus replication”, I don’t think the experiment was designed appropriately. The author test the inhibitory effect of bavachin against rotavirus in the early (6h) and late (18h) stages. For the early stage (6h) group is fine, but the virus in the late stage (18h) group was already inhibited during the first six hours. The bavachin-18h-incubation group cannot support the conclusion that bavachin can suppress viral replication in the late stage. The suitable design is that the bavachin should only be added and incubated with infected cells in the final 6h.

5. The author can consider using the GFP-carrying virus to identify when bavachin exhibits its inhibitory function during viral replication.

Comments on the Quality of English Language

 Extensive editing of English language required